# The Potential of Improving Construction Transport Time Efficiency—A Freight Forwarder Perspective

Farah Naz [1,*], Anna Fredriksson [1] and Linea Kjellsdotter Ivert [2]

1   Department of Science and Technology, Linköping University, SE-60174 Norrköping, Sweden
2   The Swedish National Road and Transport Research Institute (VTI), SE-58330 Linköping, Sweden
*   Correspondence: farah.naz@liu.se

**Abstract:** Construction transport, in general, is not carried out efficiently, resulting in unnecessary costs and $CO_2$ emissions. Although it has been found that there is a great potential to improve construction transport, little research has been conducted within this field. The purpose of this study is to contribute to the construction transport time efficiency by identifying non-value-adding activities and their causes from a freight forwarder perspective. A single case study was conducted and two flows, a goods delivery flow and a waste material flow, were mapped and analyzed with the help of value stream mapping (VSM). We ran two workshops to discuss the findings of the VSM. The results from this study show that there is large potential for improving construction transport time efficiency as over 40% of the time was used for non-value-adding activities. Although transport suffers from similar non-value-adding activities undertaken in other industries, this study identifies the transport activities with respect to construction transport. The findings further highlight areas of potential improvements, such as improved information sharing, planning, and coordination among all stakeholders. This knowledge can be used by the freight forwarder and the main contractor to improve construction transport time efficiency. The limitation of this study is that it is based on a single case of a freight forwarder. It does not provide a full picture of construction transport efficiency but rather a starting point for future studies.

**Keywords:** transport efficiency; construction transport activities; value stream mapping

## 1. Introduction

The demand for construction transport is increasing due to intense urban activity [1] and construction transport is responsible for a great amount of carbon dioxide ($CO_2$) emissions [2,3]. Earlier research estimated that, in a construction project, the transport emissions make up about 10% of the $CO_2$ emissions during the construction period [4]. Seeing that construction transport has a substantial environmental impact, there is an emerging need to address its efficiency [5]. This is especially necessary as construction transport is generally not efficiently carried out [6]. Previous studies have identified that there is a great potential for improving transport by reducing the time spent on various construction transport activities [5,7,8]. For instance, there is potential to reduce the time spent on planning, loading, unloading, and driving [5]. This can help in reducing unnecessary transport, resulting in lower $CO_2$ emissions without affecting the service level [6].

The focus on the construction transport has been growing since the early 21st century [9]. Earlier research within the field of construction logistics has focused on applying the same principles and concepts as in the urban freight transportation literature [7,9] and typically with a focus on cost and productivity aspects on a relatively abstract level [10–13]. Refs. [9,14–16] have discussed different actors' perspectives within construction transport and focused on relationship management. There are a few papers that have considered transport planning on a general level [17–20] and focused particularly on the cost and

productivity aspects within construction [10,11,13,21]. It can be seen that construction transport efficiency has not been given much attention in earlier studies, maybe because transport usually gets camouflaged by the material price [9]. Recent studies [22–24] have highlighted the importance of considering transport distances to conduct a more robust assessment of construction environmental impacts. However, this study focuses on time efficiency and its relation to environmental impact, with an emphasis on better planning and organizing for the freight forwarder in the construction supply chain. This continues the research initiated by [9] of increasing the understanding of the construction transport triad and the importance of relationships and coordination to increase transport efficiency. The need of research within the area of efficient construction transport and especially on the specific activities carried out and their relation to time efficiency is further emphasized by [25].

Time efficiency is an important part of transport efficiency [26] and, from a transport perspective, time efficiency is defined as distributing daily deliveries in the shortest possible time [25]. The focus in the paper is on the freight forwarder responsible for transport. From freight forwarder perspective, the money is made over time through the number of trips their lorries can make [27]. The better the time efficiency, the more money can be made [27]. To improve time efficiency, a necessary step is to make the construction transport efficiency gap visible [6]. Ref. [28] suggests that with the help of measuring the time spent for vehicles on different activities, transport efficiency can be estimated. Hence, to study the time spent and lost when performing different activities within the transport flow is an important task [5]. To do this, value stream mapping (VSM), a tool within lean, is used to visualize activities in a flow [29] and identify value-adding and non-value-adding activities [30].

The purpose of this study is to contribute to the construction transport time efficiency by identifying non-value-adding activities and their causes from a freight forwarder perspective.

The remainder of the paper is structured as follows: The frame of reference is presented, followed by the methods used for collecting and analyzing data. The results from the case study are presented and analyzed. Based on the theoretical framework and the empirical results, value-adding and non-value-adding activities are discussed. Finally, conclusions are drawn, and further research is suggested.

## 2. Frame of Reference

This section starts by defining transport efficiency. Next, the current understanding of construction transport is elaborated. Thereafter, the perspective on value-adding and non-value-adding activities within transport is presented. Finally, construction transport waste leading to non-value-adding time is discussed.

### 2.1. Transport Efficiency

Transport efficiency is a fuzzy concept [31]. This means that it lacks clear definition and is perceived differently by different researchers, depending upon who you ask [25]. Ref. [32] define transport efficiency as "producing a service with less resource consumption without reducing the logistics performance in terms of costs and delivery service". Ref. [33] defines efficiency "as a ratio between the outputs and the inputs of a given activity". Thus, efficiency is affected by how well the resources are being utilized [34]. In other words, efficiency is all about how to get the most out of a fixed number of resources [25].

According to [23], transport system efficiency depends on "how resources are utilized within the system. These resources include business models, vehicles, drivers, information technology, and infrastructure" [23]. It can be said that transport efficiency considers how resources are utilized, how transport activities are coordinated and how involved actors interact [9].

Time efficiency is an important part of transport efficiency and, from a transport perspective, time efficiency is defined as "distributing daily deliveries using the shortest possible time" [25]. Based on these earlier studies, transport efficiency is defined here

as moving the maximum number of goods from one point to another in as little time as possible.

Transport efficiency can be achieved by implementing technology, organizing transport in the most suitable way possible, maximizing resource utilization, coordinating transport activities, etc. [9,35]. Ref. [32] highlight that transport efficiency is not solely about technical improvements; behavioral and operational aspects are also of considerable importance. Improving the time efficiency of transport activities creates savings for the transport operator, so they can perform more value-adding activities throughout the day with the same number of resources [1]. For instance, reducing the time spent on planning, loading, unloading, and driving results in additional available time for truck drivers to perform more activities [5,36]. Thus, the transport service provider can reduce the number of trucks without affecting the service level [7]. Furthermore, it alleviates congestion and lessens the environmental impact (ibid).

### 2.2. Construction Transport

Construction transport includes "the delivery of material, machinery and equipment to construction sites, as well as the transportation of waste, soil and rock masses away from the sites" [6]. Construction transport is complex, including several types of transport such as long-distance transport, involving heavy trucks with bulky materials, medium transport, with soil and rock masses, short distance local pickup/delivery, by small trucks making frequent supply deliveries on urban roads, and return transport (vehicles containing waste, packaging and used equipment), via dump trucks [37,38].

Construction transport includes deliveries that vary from small to large, between packages, palletized goods, framing materials and bulk materials, etc. [39]. The types of transport required vary during the project; early phases of construction require heavy transport because of the use/removal of concrete and other rock masses, whereas later construction phases consist of small packages and pallets [6]. Furthermore, the waste flow cannot be ignored [6] and in fact the construction industry is one of the largest producers of physical waste and recyclable materials [40].

To bring improvement within construction transport efficiency is hard due to the way construction industry is organized [9]. Construction productions involve multiple stakeholders and is performed on a project-by-project basis in temporary organizations [21,41,42]. Ref. [34] suggest that the root cause of inefficient construction transport is a lack of transport planning and organizing among the involved actors, such as contractors, sub-contractors, material suppliers and transporters. This scenario, where each actor follows its "best" coordination logic, results in the lack of joint coordination [9,34]. Furthermore, construction transport is often ignored in business agreements among different parties [9]. Transport is usually included in the material price; thus, the control of transport planning remains with the suppliers [9]. All these factors together make it hard to notice inefficiencies that exist within construction transport [43].

### 2.3. Value-Adding and Non-Value-Adding Activities

To identify value-adding and non-value-adding activities, a lean tool known as value stream mapping (VSM) is commonly used [30]. It uses a flow chart approach documenting every step/activity in the process [30]. VSM is a unique recording technique as it captures detailed information, such as working time, process time, and the handling time of a particular activity, and allows us to gain a good understanding of complete flows.

As per [7], the activities valued by customers and have a direct positive impact on the final product are considered as value-adding activities, whereas non-value-adding activities do not play any role in enhancing the value of the final product and rather have a negative impact. Ref. [44] viewpoint is that although all activities incur cost and consume time, these are value-adding activities that add value to the processes. This is why a lean approach aims at redesigning the processes to achieve two main goals: (1) remove or reduce the number of non-value-adding activities; and (2) spend more time on value-adding activities.

Value-adding and non-value-adding activities are defined from a customer perspective; however, from a producer perspective, the term waste is also used [7]. Ref. [45] suggest that there are three main categories in which waste is classified, which are "material waste, time loss and value loss. Material waste is related to the transformation perspective (non-optimal use of material and non-optimal use of machinery, energy, or labor) [45]. Time loss is related to the flow perspective (unnecessary movement of people, unnecessary work, inefficient work, waiting, space not being worked in, materials not being processed and unnecessary transportation of material) [45,46]. Value loss is related to the value perspective (lack of quality, lack of intended use, harmful emissions, and injuries)" [45]. The focus in this study is on time loss, with the aim to identify and eliminate non-value-adding activities to improve the overall efficiency of company's operations.

*2.4. Non-Value-Adding Activities in Construction Transport*

The construction industry has been suffering from inefficiency, which is often studied within the research stream of lean construction management [47]. Lean construction management typically focuses on the construction site and traditional construction activities and not on the transport to and from the construction site. For instance, it has been found that transportation is a kind of interruption that should be avoided to minimize waste in construction [45]. Transport activity is defined as "an activity that is not directly involved in construction but is required to facilitate it" [48]. Though, there exists uncertainty as to what are value-adding and non-value-adding activities within construction transport. Ref. [49] suggest that too many transport activities should not be eliminated as this may negatively affect the efficiency of the construction process.

It becomes clear that construction transport is seen more as a non-value-adding activity, referred to as "waste" in previous studies on lean construction management [46,50–52]. Refs. [45,46] state that transportation is related to the unnecessary movement of people or unnecessary transportation of materials. Furthermore, Ref. [53] emphasize that transportation lacks efficiency due to the use of equipment, materials, labor, and capital in quantities greater than those necessary for construction. Ref. [54] also focused on the transport of waste on the site, such as access/movement, adjustment of components, working area, storage of materials or components, equipment/tools, etc. Ref. [55] consider idleness of time as waste.

Thus, the earlier research on lean management for construction has not provided much information necessary for this study. Still, the causes of transport "waste" at the construction site presented in previous research are still valid. Ref. [53] defined the main causes of transportation waste:

- Access/mobility problems: Any kind of route obstruction that makes the transport activity difficult.
- Storage: Inappropriate space for material storage or material stored in an inappropriate manner.
- Equipment: Unavailable, damaged, or inappropriate equipment for transportation, generating the adaptation of other equipment for the transportation or appropriate equipment, but used in an inappropriate manner.
- Workforce: Insufficient number of workers to perform the transportation activity.
- Packaging material: Poor packaging conditions of the material, which make the transportation slow and difficult.
- Information: Lack of necessary information for the employees for high-quality transportation performance.

These definitions of construction site transport waste are similar to the transport waste covered in the transport research [56]. Ref. [57] suggest that road operations can become efficient by avoiding extra travel, for which it is crucial to determine exact transport distances [24]. Better planning of transportation and delivery schedules can also help to achieve efficiency [58]. Transport planning in construction depends upon the planning of the construction project [8]. The construction transport vehicle unloads the material upon

reaching the construction site and picks up/loads waste from the site [6]. This allows the construction site to have enough space for performing the tasks productively. Over the last few decades, the construction industry has started to implement just-in-time (JIT) system to avoid on-site material accumulation [12]. To ensure the arrival of JIT material, it is important to coordinate the planning of construction transport with the planning of the construction activities. This will reduce the risk of delays that otherwise lead to non-value-adding waiting time for crew, thus affecting the efficiency of construction projects [58]. However, if not properly coordinated, there will be non-value-adding time for the freight forwarders.

According to [7], insufficient planning leads to further decreases in the overall operations performance. He further adds that unnecessary waiting at the time of loading and unloading occurs because of queues, limited access to forklifts and other equipment. Furthermore, incorrect routes make drivers drive unnecessarily long distances, either due to incorrect address information or because of route selection being based on the driver's personal preference. [7] suggests that unnecessary movements result due to confusion regarding the loading or unloading areas in case of the absence of designated loading and unloading zones. This makes drivers drive further and creates disturbances. Ref. [7] found that damages during transport increase administrative tasks and extra movement.

*2.5. Value Stream Mapping*

Value stream mapping (VSM) is a useful technique for recording processes, identifying waste and making recommendations for action [59]. A value stream map visually shows all critical steps in a specific process and easily quantifies the time taken at each step using a flow chart approach [30]. It has the goal of analyzing and optimizing the entire process [29] and helps in visualizing the current gaps [60] to propose a more efficient future state for any process [30]. VSM consists of three important steps [30]:

(1)  Creation of the current process state
(2)  Identification of inefficiencies in the process using lean principles
(3)  Development and suggestion of action plans to achieve the proposed improved future state

An important part of value stream mapping is classifying activities as value-adding or non-value-adding based on time as the non-value-adding activities contribute to additional time consumption [60].

VSM has been used extensively in the manufacturing area [61,62]. However, it has also been used in other areas such as logistics and supply chain, healthcare, and other service operations [61].

According to [61,62], at the supply chain level, in addition to overproduction due to poor information flow, unnecessary inventories and transportation have become large sources of waste that need to be identified and eliminated. Value stream maps generally provide a good amount of information, such as the frequency of trips, distance travelled, shipping size, defective rate, etc. [29]. A value stream map for transportation is shown below in Figure 1 as an example [61].

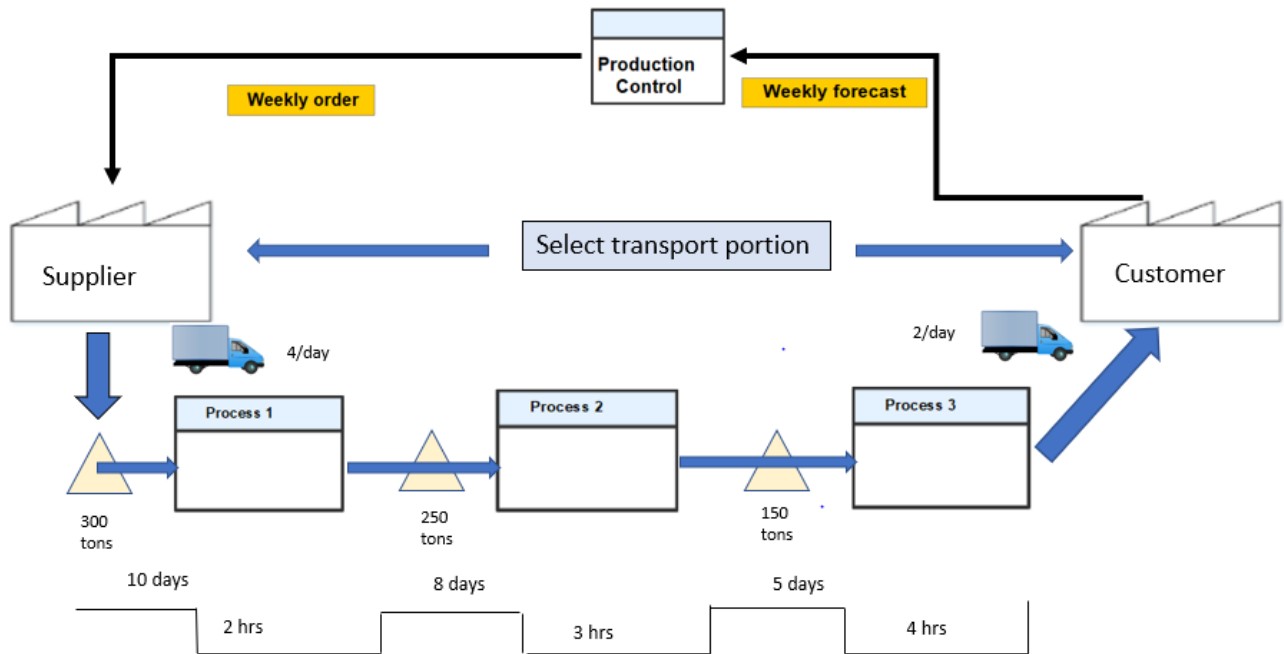

**Figure 1.** Value stream map of transport in supply chain (adapted from [61]).

## 3. Material and Method

As there are few previous studies on construction transport efficiency, a case study was used in this research. A case study approach offers the opportunity to dig deep into various phenomena [63] and this study is based on a subjectivist standpoint, where understanding of the context is very important. A case study is a favorable methodology for studying a phenomenon in its context [63]. Furthermore, the value stream mapping (VSM) technique has been used as an organizing guide to collect data. VSM has been chosen because it is the most common method of studying value-adding and non-value-adding activities [30]. Besides this, VSM allows one to make in-depth observations of the phenomenon at hand [29].

### 3.1. Case Selection

The selected case is a regionally major freight forwarder in Sweden, with a vision to become the most sustainable and efficient transport provider. The company´s diverse portfolio made it possible to study two different flows: (1) goods delivery flow and (2) waste material flow. Besides this, accessibility was also one of the factors in selecting this case as the VSM approach requires frequent contact and trust.

The case company perform construction logistics and want to push construction transport towards sustainability. It consists of 118 employees and owns 159 vehicles. All of the company´s vehicles run on renewable fuel. The company offers everything, from environmental consulting services, waste and recycling solutions, construction logistics, crane truck and machine services, other related services such as goods handling, storage, property management, etc. The company is currently lacking more long-term transport plans, and the planning of deliveries or pick-ups takes place one day in advance of the activity. The focus for the future is to improve long-term planning and to prepare a "weekly order list". Presently, the company puts little focus on using tools to improve routings and puts trust in the drivers' ability to plan their own routes and work. The drivers are currently facing challenges in terms of finding the right addresses and unloading zones and there are long and complex information chains due to the involvement of multiple stakeholders. Due to this, a lot of time is consumed in making phone calls, manual copying, and checking documents again and again, which could have been spent actively

on planning and organizing transport operations. Making 12 deliveries in a day is then considered efficient, although the number of deliveries is dependent on their length.

### 3.2. Data Collection

Two transport flows, a goods delivery flow and a waste material flow, were studied from a time efficiency perspective. Data were collected by conducting a time study. The activities and timings for each activity were recorded. According to [64], a time study is relatively simple in nature, providing a short learning curve. Furthermore, a time study is a work measurement technique for recording the times and rates of working for the elements of specified job carried out under specific conditions [64]. It is also a technique used to analyze data to determine the time necessary for carrying out the job at a defined level of performance [64]. By recording times for different activities, it can be observed how time varies when performing activities due to the direct and indirect deviations caused by fatigue, other personal circumstances, and unavoidable delays [5]. The time study was performed by joining a truck driver for a full day (8:00 am to 5:00 pm) on two different occasions. Pictures of unloading zones at construction sites and the material being loaded and unloaded were taken. Google Maps was used to identify the routes and delivery locations.

To find out more about how deliveries were planned and managed within the studied company, an interview was conducted with the business unit manager. The interview was semi-structured and recorded. Interview transcripts were sent back to the interviewees to increase reliability. The interview guide is presented in Appendix A. Two workshops were also run on two different occasions: one for a time period of one hour with 7 participants. The main objective of conducting this workshop was to discuss the data gathered during the time studies. The other workshop was conducted for a time period of two hours, consisting of 14 participants. The main goal of this workshop was to discuss the findings of the VSM. The participants of both workshops were construction professionals working in logistics.

### 3.3. Data Analysis

The units of analysis were value-adding and non-value-adding time. The analysis was performed by calculating the time for each activity. Based on the work conducted by [49], a taxonomy to describe non-value-adding, non-value-adding but necessary and value-adding transport activities in construction transport was developed. According to [49], transport activities within construction can be categorized as: (1) productive activities that add value to the final product (in this study, these are referred as value-added activities); (2) contributory activities that do not add value in a direct way to the final product but are necessary activities to be performed (in this study, these are referred as necessary but non-value-adding activities); (3) non-productive activities that do not add value and are considered as a waste of time and classified as avoidable, unavoidable and idle (in this study, the aforementioned activities are referred as non-value-adding activities).

The categorization of the activities was conducted based on: (1) the importance of a particular activity in a transport flow; (2) time loss vs. benefit when performing an activity; and (3) alternative methods of performing an activity. This is also based on findings from previous studies, such as [1,7,44,45,47].

The time for value-adding activities was summed up to calculate the total value-adding time and time for non-value-adding activities has been summed up to calculate total non-value-adding time. The percentages of value-adding time and non-value-adding time were calculated based on total delivery time.

The validity of this study was ensured by collecting data via observations and then confirming interviews and workshops. Construct validity was assured by defining the terms throughout. Reliability is achieved by ensuring that the data collected were well documented.

Table 1 presents the six different deliveries in terms of time and more detailed description.

Table 1. Summary of goods delivery flow.

| Deliveries | Description | Time Taken |
| --- | --- | --- |
| **Delivery 1 (Renovation materials)** | Loading took place after picklist being collected. Truck departed at 6:58. Three stops were made to search for the right address. Before delivering, truck waited for 16 min at wrong parking place. **Waste:** Collection of picklist, searching for the right address, searching for the unloading space, contacting the concerned person, waiting, travelling back empty, taking picture of unloaded material | 6:53–7:42 (49 min) Total Delivery time = 49 min Value-adding time = 16 min Non-value-adding time = 33 min |
| **Delivery 2 Concrete Slabs** | (a) Picklist got collected after discussion regarding addresses. Loading was carried out. Truck arrived at the wrong address. After contacting the concerned person, delivery was made at the right address. (b) Another delivery was made at a site with no proper unloading area. Delivery was made through a very narrow passage. Because of improper unloading space, more time got consumed along with enhanced risk of vehicle damage. **Waste:** Collecting picklists, reaching at the wrong address, contacting the concerned person, difficult access to unloading space, taking pictures of unloading material, travel back empty | 7:45–8:16 (31 min) 8:17–8:55 (38 min) Total delivery time = 69 min Value-adding time = 49 min Non-value-adding time = 20 min |
| **Delivery 3 Construction material** | It took 4 min to collect the invoice. The route was rough and not easy to find the way. It took 8 min to unload. Truck travelled back empty. **Waste:** Waiting for the picklist to get confirmed, taking picture of unloaded material, travel back empty | 9:17–10:09 (52 min) Total delivery time = 52 min Value-adding time= 28 minNon-value-adding time= 24 min |
| **Delivery 4 Construction material** | Delivery 4 went smooth. It was easy to find the address. **Waste:** Taking pictures of unloaded material and travel back empty | 10:10–10:34 (24 min) Total delivery time = 24 min Value-adding time = 16 min Non-value-adding time = 8 min |
| **Delivery 5 Construction material** | It took time to unload the loaded material because of delicate nature of iron rods which were long, thin, and heavy at the same time. It required a lot of skill to handle and offload the material. **Waste:** Truck travelling back empty. | 10:37–10:55 (18 min) Total delivery time = 18 minValue-adding time = 12 min Non-value-adding time = 6 min |
| **Delivery 6 Bandage boxes and construction material** | (a) Delivery was made just to deliver four small bandage boxes. (b) Delivery was made to the adjacent city and it went smooth. **Waste:** Collection of picklists, delivered 4 small bandage boxes, travelled back empty, taking pictures of unloaded material in delivery b. | (a) 10:56–11:17 (21 min) (b) 11:18–12:30 (72 min) Lunch break 12:37–1:39. Reached back by 2:33. Travel time to reach back (66 min) Total delivery time = 159 min Value-adding time = 89 min Non-value-adding time = 70 min |

Table 2 presents the six different dumpings in terms of time and more detailed description.

**Table 2.** Summary of waste material flow.

| Dumping | Description | Time Taken |
|---|---|---|
| **Dumping 1 Waste material** | Filled waste container was loaded from construction site. Net was put on it. After reaching at recycling facility, filled waste container got weighed, receipt of which got collected. A stop was made before dumping to remove net. After dumping, empty container was weighed again. Dump truck travelled back to same construction site to put the empty container there. Time spent waiting due to presence of another truck. **Waste:** Too much stopping at recycling facility, i.e., for weighing and collecting paper receipt, for removing net and finally for dumping. 4 min waiting to put the empty container back at site because of another truck delivery. | 7:15–7:58 (43 min) Total time = 43 min Value-adding time = 26 min Non-value-adding = 17 min |
| **Dumping 2 Waste material** | Waiting before loading the filled waste container. After reaching recycling facility, container got weighed both filled and empty. After dumping, truck travelled all the way back to put the container. **Waste:** Waited for 4 min to enter the site for lifting up the filled container. | 8:02–8:38 (36 min) Total time = 36 min Value-adding time = 21 min Non-value-adding = 15 min |
| **Dumping 3 Waste material** | For dumping 3, driver reached the construction site by 8:59. The truck was parked close to the filled container so that it can be loaded easily. After loading, net was put and then driver went for dumping and after dumping travelled back with the empty container to put it back at the construction site from where it was picked up. **Waste:** Parking the truck so that the filled waste container can be loaded easily. Putting and taking off the net 20 min waiting time to load the filled container from construction site. Net was put on after loading. At recycling facility, truck got weighed with filled | 8:50–9:28 (38 min) Total time = 38 minValue-adding time = 25 min Non-value-adding = 13 min |
| **Dumping 4 Waste material** | container. Net was removed before dumping and afterwards weighing was carried out again. At this recycling facility, inspection was made before dumping. Empty container was put back. **Waste:** 16 min waiting to lift up the filled waste container, one additional stop was made due to inspection at the recycling facility | 9:28–10:42 (74 min) Total time = 74 min Value-adding time = 35 min Non-value-adding = 39 min |
| **Dumping 5 Waste material** | At 11:40 the driver was at the construction site and opened the gate by himself. At 11:50, driver parked the truck to lift up the container. At 11:51 full container was ready to be picked up. At 11: 54 lifting was carried out and by 11:55 after putting the net, the driver also closed the site gate. By 12:13, driver reached the recycling facility and after weighing and removing the net, the material got dumped. After dumping, driver travelled back to the site again from where the container was picked up to put the empty container back. **Waste:** One stop was made to open the gate of construction site | 11:40–12:45 (65 min) Total time = 65 min Value-adding time = 38 min Non-value-adding = 27 min |
| **Dumping 6 Waste material** | After a lunch break, filled container was lifted up and dumped at the recycling facility. **Waste:** Collecting paper weighing receipt, putting and taking off net | 1:03–1:42 (39 min) Total time = 39 min Value-adding time = 25 min Non-value-adding = 14 min |

## 4. Results

With the help of the empirical data presented in Tables 1 and 2, the following value stream maps for goods delivery flow and waste material flow were developed.

### 4.1. Value Stream Mapping—Goods Delivery Flow

The goods delivery flow consists of ten activities, presented with the time taken in Figure 2. All these activities are classified into three categories, i.e., value-adding, non-value-adding and necessary but non-value-adding. The times in red in Figure 2 show waste, whereas the time in black shows effective time utilization. The time presented at the end of each row shows the total time taken (for example, delivery 1 took 49 min to complete).

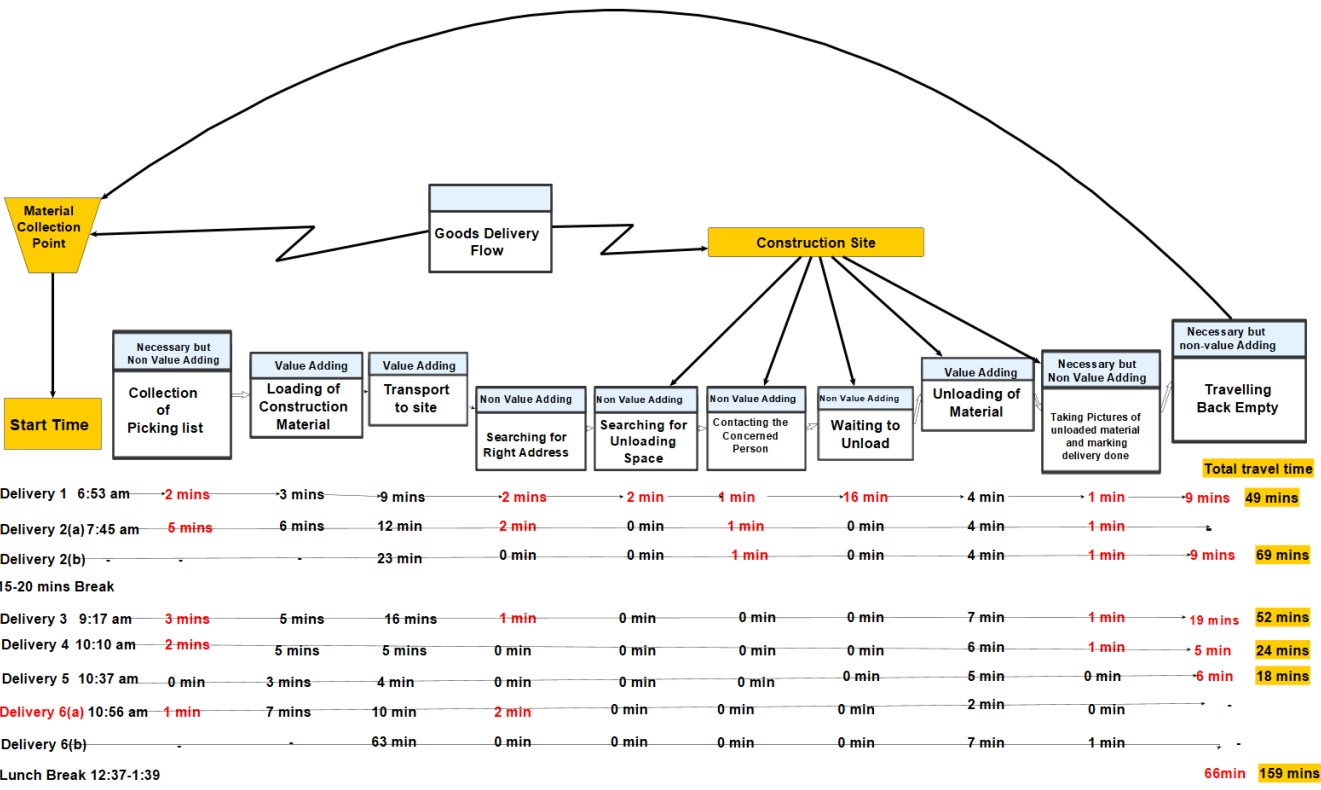

**Figure 2.** Value stream mapping of goods delivery flow.

The loading, transporting, and unloading of construction material are classified as **value-adding activities** because without these activities the whole purpose of goods delivery flow cannot be fulfilled (i.e., the movement of construction material from the supplier to construction site).

Searching for the right address, searching for unloading space, contacting the concerned person, and waiting to unload are classified as **non-value-adding activities** because all of these activities should not be needed and could be avoided through improved information sharing and planning.

The collection of picking lists, taking pictures of unloaded material, marking delivery to be carried out, and travelling back empty are classified as **necessary but non-value-adding activities** because these are important but can be significantly shortened through digitalization, improved planning, and information sharing.

### 4.2. Value Stream Mapping—Waste Material Flow

The waste material flow consists of ten activities presented with the time taken in Figure 3. All these activities are classified into three categories, i.e., value-adding, non-value-adding and necessary but non-value-adding.

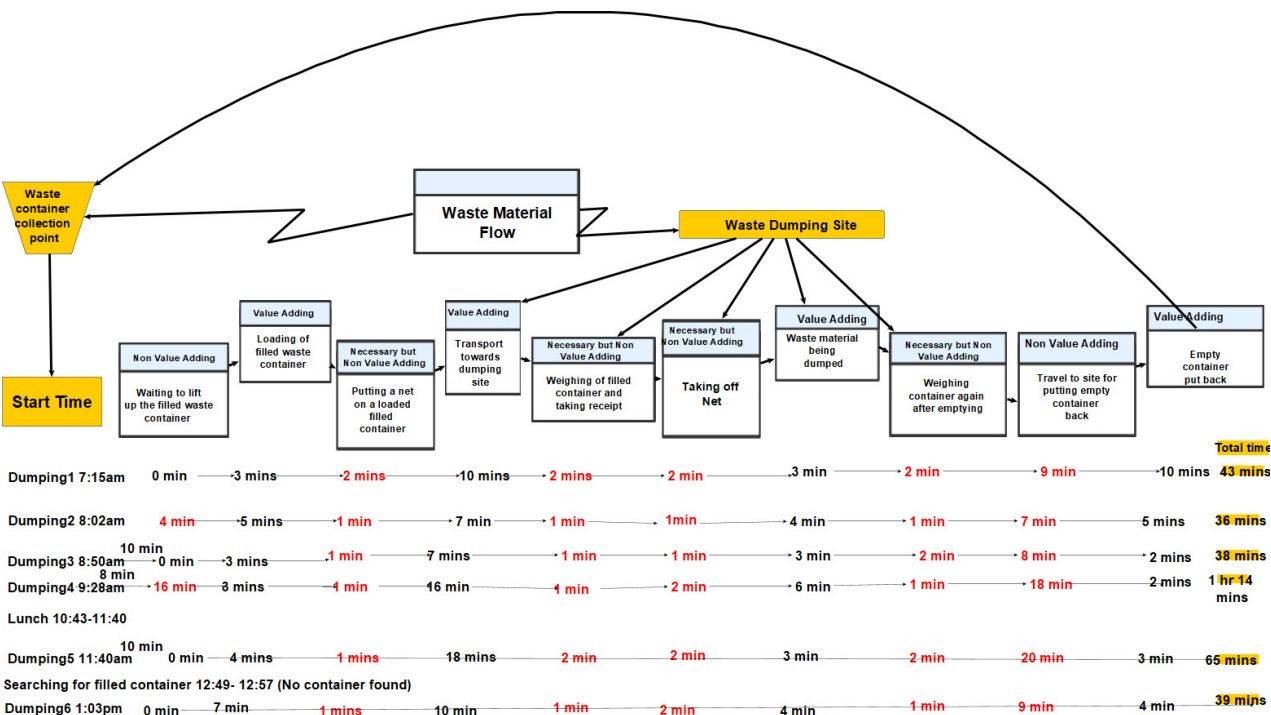

**Figure 3.** Value stream mapping of the waste material flow.

The loading of a filled waste container, transport towards the recycling facility, and waste material being dumped are considered as **value-adding activities** because without these activities the whole purpose of return flow cannot be fulfilled (i.e., the movement of waste material for either disposal or re-use from construction site to recycling facility). Returning the empty container back to the construction site for refill is also considered as value-adding because it is important to do this so that construction personnel can throw waste in it.

Waiting to lift up the filled waste container and travelling back to site only for putting the empty container are classified as **non-value-adding activities** because these activities should not be needed and could be avoided through improved planning and information sharing or better vehicle design for the latter so that vehicles can take two containers simultaneously (one filled and one empty within one trip).

Putting a net on a loaded filled container, weighing of a filled waste container, and taking a paper receipt, taking off the net from the filled container before dumping, weighing the container again after dumping and collecting a paper receipt are classified as **necessary but non-value-adding activities** because these are important but can be avoided by digitalization and automation.

The times in red in Figure 3 show waste, whereas the time in black represents effective time utilization. The time shown at the end of each delivery represents the total time taken for that delivery. The times shown above the arrows at the beginning of dumpings 3, 4 and 5 represent the time taken to reach the site from where the filled waste container is to be lifted. This has not been mentioned for other dumpings because it takes less time to reach the site to lift the filled waste containers.

The time taken by the truck to come from and to the parking area at the start and end of the day has not been considered here.

## 4.3. Summary of Results

### 4.3.1. Goods Delivery Flow

Figure 4 shows that the total time taken for full day delivery was 371 min, where the value-adding time was 210 min, corresponding to 56.6% of the total time, and the

non-value-adding time, which includes necessary but non-value-adding, time was 161 min, corresponding to 43.4% of the total time.

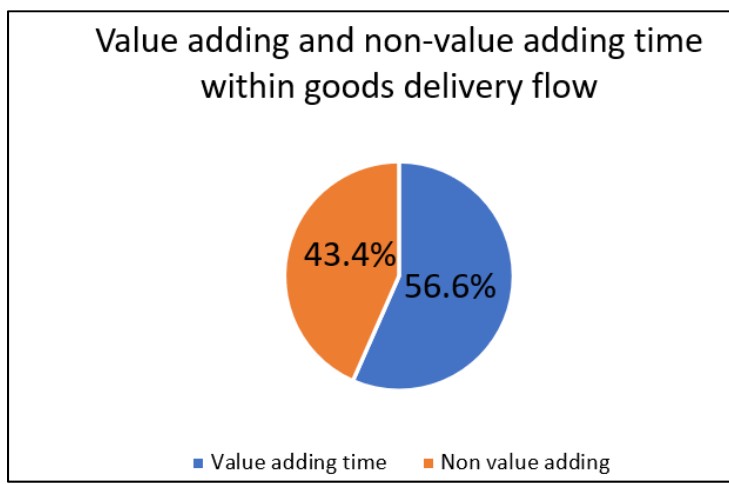

**Figure 4.** Value-adding and non-value-adding time (includes necessary non-value-adding time) within goods delivery flow.

### 4.3.2. Waste Material Flow

Figure 5 shows that the total time taken for full day dumping was 295 min, where the value-adding time was 170 min, corresponding to 57.6% of the total time, and the non-value-adding time, which includes necessary but non-value-adding time, was 125 min, corresponding to 42.4% of the total time.

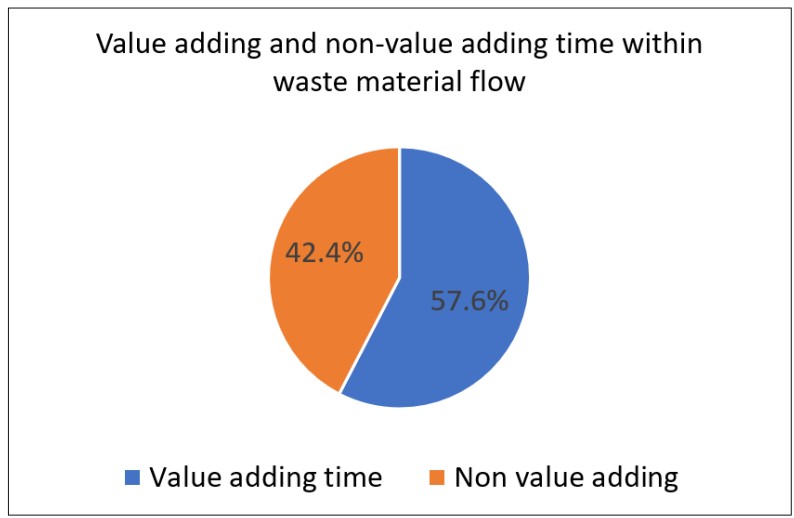

**Figure 5.** Value-adding and non-value-adding time (includes necessary non-value-adding time) within waste material flow.

### 5. Discussions

The findings of this study show that almost 40% of the time is consumed performing non-value-adding activities, mainly waiting to collect the picklist, loading and unloading, searching for the right address, searching for the unloading space at the construction site, contacting the concerned person, taking paper receipts, and travelling empty. The causes behind the many non-value-adding activities, and thereby low efficiency, have all been brought up in previous studies, such as a lack of planning [43,65,66], lack of communication and information sharing [65,67–69], and multiple stakeholders, resulting in long and complex information chains [70,71], and lack of digitalization [72]. This highlights the need to use ICT tools and digitalization for improved planning, communication, and

co-ordination which in turn will decrease the time spent on non-value-adding activities. The role of the use of ICT tools, digitalization, and technology in transport planning, communication, and coordination is discussed in Table 3.

**Table 3.** Proposed solutions and their impact on construction transport planning, communication and coordination.

| Proposed Solution | Impact |
|---|---|
| Use of ICT tools (advanced mapping devices, telematics, GPS technology, integrated software, real-time information tools, etc.) | The use of ICT tools will help improve information sharing which will lead to better transport planning and decision making. Furthermore, this will improve communication as well as coordination. The ICT tools will enable connection among construction sites, drivers, and terminals, resulting in traceable transport. The use of advance mapping tools will help the drivers to plan their routes in a better way and it will provide them information regarding road obstacles in advance, reducing congestion in urban areas. Additionally, this will allow route optimization, resulting in less empty transport. Ref. [7] suggests that use of ICT tools improve loading and unloading process by sending pre-arrival notification which will help personnel to be prepared and make the equipment ready for the loading and unloading. Waiting due to bottlenecks in terminals will be reduced with the use of ICT tools [7] |
| Digitalization (e-mails, electronic receipts) | The digitalization will reduce drivers' administration through the use of electronic documentations such as digital receipts, e-mails, etc. The reliance on paper invoices and receipts and manual data recording is time consuming as well as error prone. With digitalization, waiting due to paper-based administration will be reduced. |
| Technology (Smart waste containers, special vehicles, etc.) | The sensors used in smart waste containers will notify the driver that the waste container is filled and is ready to be picked up. This can reduce the number of trips that the driver currently makes just to check whether the container is filled with waste and ready to pick up or not. Furthermore, it has been observed that even when delivering small boxes to construction sites a big vehicle is being used. This suggests that there should be special vehicles according to the type and size of material. This will increase overall transport efficiency. In addition to this, it has been observed that to load the filled waste container, the driver has to make quite a few adjustments to align the truck and the container. It is suggested that with the help of some magnetic technology, the container can be made aligned with the truck by itself. This will reduce the excessive number of unnecessary movements as well as reduce driver´s frustration. |

With the help of these suggested solutions, transport planning, communication and coordination will be improved resulting in less time being spent on non-value-adding activities, which will in turn improve construction transport time efficiency as well as reduce $CO_2$ emissions. For example, in the case of goods delivery flow by improved transport planning, communication and coordination, the total time taken in all activities will be reduced from 371 min to 325 min, which is a reduction of 46 min, as shown in Figure 6. This much time will get reduced for all six deliveries. By extrapolating this reduction of 46 min to per week or per month deliveries, the time as well as environmental impact can be presumed. Although the non-value-adding time for travelling back empty is not considered here since the truck has to come back even though it is filled. Here, travelling back empty is shown as necessary but non-value-adding because coming back empty is what makes it non-value-adding.

This study contributes with a comprehensive, empirically based picture of non-value-adding activities in construction transport. Previous literature from transport research area [1,7] has shown that non-value-adding activities are mainly those related to waiting and unnecessary vehicle movements. Based on the findings of this study, it can be said that this is also true for construction transport; i.e., the non-value-adding activities are waiting to load and unload, searching for the right address and unloading zones, un-

necessary movements such as unnecessary stopping for putting net and taking off net, and weighing of filled waste container and empty waste container. By looking in detail at transport-related activities enabled through the VSM approach, this study highlights earlier unnoticed activities, specifically related to construction transport, such as putting the net one, taking off the net, weighing the truck twice (filled and empty) in the waste material flow and searching for the right addresses, finding unloading spaces, and contacting the concerned person in goods delivery flow, among other prominent activities such as loading and unloading. The reason that these activities have not been highlighted in earlier transport research by, for example, Ref. [7] is that these activities to the main part are construction flow specific. Additionally, the reason that they have not been highlighted by construction logistics researchers (for instance, Refs. [9,17], is that they have focused on the relationships within the supply chains and not the specific activities. However, to increase the understanding of construction transport, it is essential to know how efficiency can be improved.

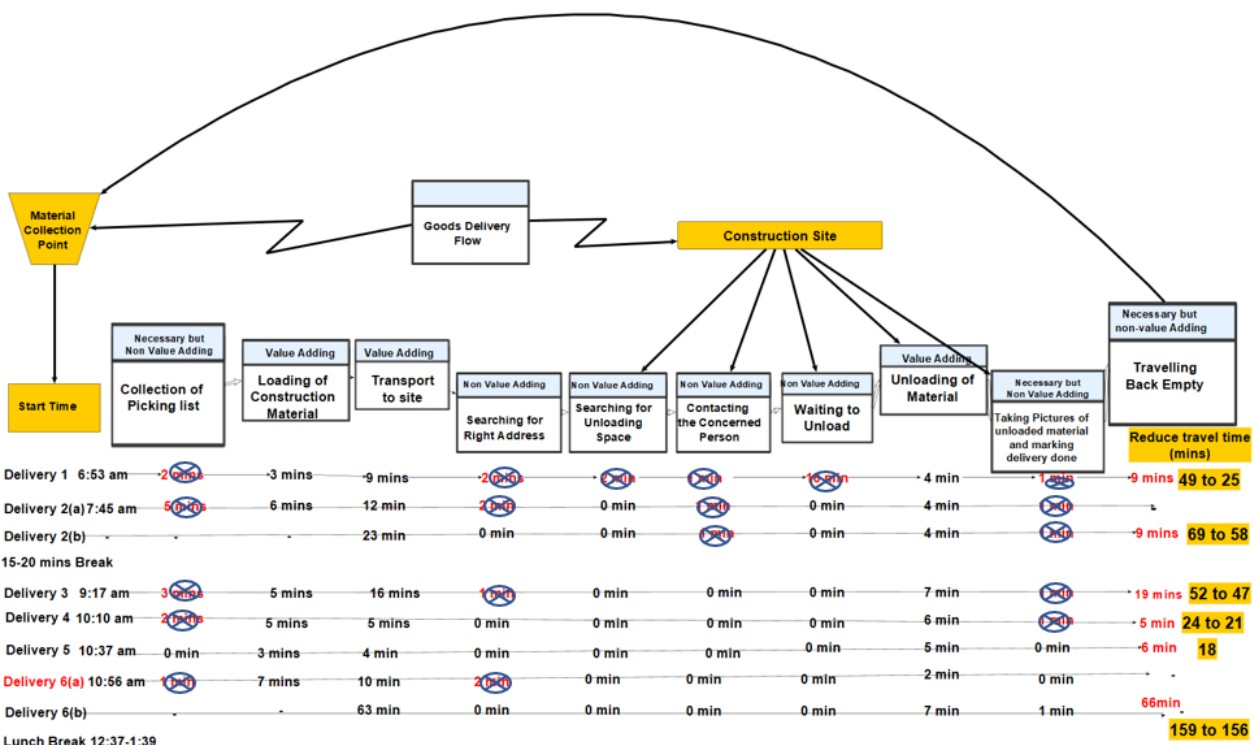

**Figure 6.** Future state of goods delivery flow.

Another issue specific to construction transport is the length of the trip. Local material transport with rather short distances have remained the scope of this study. Compared to many other transport efficiency studies, this shows the need to highlight the issue of the length of the trip. Because the longer the trip, the more value it will add as compared to a shorter trip since the value-adding activities are loading, transporting, and unloading. The importance of transport distance has also been highlighted by [22,24,73] for measuring environmental impacts.

However, the contribution of this study is the outlining of where in the transport flow these wastes occur and how they impact the freight forwarder.

## 6. Conclusions

The purpose of this study is to contribute to the efficiency of construction transport by identifying non-value-adding activities and their causes from a freight forwarder perspective.

It fulfils this purpose mainly by identifying the transport activities with respect to construction transport and the time being spent on performing value-adding and non-value-adding activities.

The findings of this study highlight that there is a need for better planning, communication, and coordination within construction transport, and this can be achieved with the help of digitalization, the use of ICT tools, and improved technology. This will help in improving construction transport time efficiency. There will be less time spent on non-value-adding activities such as waiting, searching for the right address to unload the material, contacting the concerned person, empty travelling, etc. Likewise, there will be fewer $CO_2$ emissions because unnecessary transport will be reduced. For example, it has been observed in this study that during waiting the driver did not switch off the engine and waiting occurred for 20 min. Therefore, it is essential to plan and coordinate transport in a better way so that the waiting times can be reduced along with other non-value-adding time.

This study fills the aforementioned research gap as less attention has been paid to construction transport previously and this could be due to the fact that construction transport is often camouflaged in the material price [9]. Nevertheless, a lot of research studies have been focused on freight transport. Since the dynamics of construction transport are different than freight transport such as relatively heavy transport material, use of cranes and special equipment for loading and unloading, involvement of construction sites which are usually congested with limited space and also varies geographically. Therefore, it is necessary to understand construction transport to ensure time efficiency in the construction industry. The main contributions of the study are:

- To classify construction transport activities as value-adding, non-value-adding and necessary but non-value-adding
- To identify the time being spent performing value-adding activities and non-value-adding activities
- To propose solutions to make construction transport time-efficient

According to [54], a lean approach helps in achieving two goals, i.e., (1) the removal or minimization of the share of non-value-adding activities and (2) the increase in the time spent on value-adding activities. In order to achieve the second goal and based on the findings mentioned above, there is a need to conduct further studies to identify whether the value-adding activities are performed efficiently or not.

The limitation of the study is that it is based on a single case of a freight forwarder and only two days of observations. However, as this study was exploratory, with the aim of identifying value-adding and non-value-adding activities, the focus has been on identifying areas for further research and improvement potential, for example, to set benchmarks for each activity in terms of time and study variations among activities, etc. In other words, this study does not provide a full picture of construction transport efficiency but provides a starting point for future studies.

**Author Contributions:** F.N. is the main author, responsible for data collection and data analysis, whereas all authors (F.N., A.F. and L.K.I.) equally took part in the planning and writing procedure. F.N., A.F. and L.K.I. All authors have read and agreed to the published version of the manuscript.

**Funding:** This study has received funding from Trafikverket (Swedish Transport Administration) via a project Triple F (Fossil Free Freight). The grant number is 2019.2.2.5.

**Data Availability Statement:** Not applicable.

**Acknowledgments:** The authors would like to extend their gratitude to Triple F, the Transport Administration´s research and innovation for their initiative towards the transition to fossil free freight transport as well as for the funding of this project.

**Conflicts of Interest:** The authors declare no conflict of interest.

## Appendix A. Interview Guide

Q1. Which among the following activities do you consider as non-value adding i.e., activities that are just there and do not add much value to the whole process;

 a. Picking lists collection
 b. Taking pictures of the unloaded material
 c. Searching for filled waste container at site
 d. Weighing of filled waste container
 e. Receiving paper receipt for weighing
 f. Putting and taking off net
  Follow up question:
  Which do think can be digitalized/automated?
  Why is it important putting the net on the waste filled container?

Q2. What do you consider normal time for loading and unloading?

 a. Do you have a normal time part of the planning?
 b. Do you think that 4 to 5 min for loading and unloading is good time utilization?

Q3. How do you plan deliveries at your company?

 a. In your opinion, do flaws/shortcomings/faults/inconsistencies exist in planning and organization of deliveries?
 b. How can planning and organization of deliveries be improved in your opinion?
 c. Is it possible to plan better routing so that time is utilized efficiently i.e., to meet several delivery points in one route rather than coming and going back and forth?

Q4. Would it be possible to collect all the delivery invoices at the same time for all the deliveries planned for a day rather than collecting invoices one by one or delivery by delivery?

Q5. What challenges do you face in getting the information about unloading place/area/zone from the customer?

 a. How important do you think it is to communicate unloading space information with the driver?
 b. Do you think unloading zone information can be mentioned somewhere on the invoice so that driver has an idea, and he/she should not waste time in searching the unloading zone while being at the site?
 c. Would it be better for your company to book time with the construction site so that there is no waiting in performing loading/unloading at site and less congestion on road because waiting has been observed in some deliveries because of another in progress delivery?

Q6. Normally it has been observed that the driver after dumping the filled waste container go all the way back to put the empty container at the same place from where it was collected. Do you think there can be any alternative to it such as leaving empty containers at recycling facility and some other big truck can put all empty containers back at the site in the form of consolidated delivery?

Q7. Do you think google maps is effective in finding the right addresses? Because sometimes google map does not show the diverted routes? Is it possible to have our own maps or some other alternative?

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
