# Peer review of "The Potential of Improving Construction Transport Time Efficiency—A Freight Forwarder Perspective"

_sustainability, doi:10.3390/su141710491_

Round 1

Reviewer 1 Report

Reviewer Comments:

General comments:

It is my pleasure to review the paper entitled “Improving Construction Transport A Freight Forwarder Perspective”. This is interesting and study has been conducted and two flows, the goods delivery flow and the waste material flow have 13 been mapped and analysed with the help of value stream mapping (VSM).  The paper is within the scope of Sustainability. However, it needs some major changes before publication.

I present the following comments that can help to improve the paper:

Please check the title and revise it. 

Introduction

  1. Line 37-38 Reference?
  2. Line 46-47. Kindly extend the Introduction part by providing recent references from recent articles in the introduction section. i.e. https://doi.org/10.1016/j.jclepro.2021.130083; https://doi.org/10.1016/j.jobe.2021.103935; https://doi.org/10.1016/B978-0-12-821730-6.00028-0.
  3.  What are your research questions and hypotheses? What are the important findings expected by the readers? What are research gaps in the past, and what are your contributions to improve them?
  4. I would suggest the authors to draw some figures of results so that reader could clearly understand them. 
  5. How you have improved the Construction Transport and reduce CO2 emissions, please elaborate
  6.  

Best, 

Author Response

Dear Reviewer, 

Thank you so much for your time regarding reviewing the paper and providing very relevant, useful and insightful feedback. I have revised the draft as per your feedback. Please see the attachment for responses to the given comments. 

Reviewer 1 comments

Response

Please check the title and revise it

Thanks for the suggestion. The title has been revised from “Improving Construction Transport- A Freight Forwarder Perspective” to “The potential of improving construction transport time efficiency- A freight forwarder perspective”

Line 37-38 reference?

Reference to line 37-38 has been added. Due to additional text, the line 37-38 has become line 47-48.

Line 46-47. Kindly extend the Introduction part by providing recent references from recent articles in the introduction section. i.e. https://doi.org/10.1016/j.jclepro.2021.130083; https://doi.org/10.1016/j.jobe.2021.103935; https://doi.org/10.1016/B978-0-12-821730-6.00028-0.

Thanks for providing links to the recent articles. The reference of these articles has been added in the second paragraph of introduction chapter from line 53-58, frame of reference (section 2.4), and in the end of discussion chapter.    

What are your research questions and hypotheses? What are the important findings expected by the readers? What are research gaps in the past, and what are your contributions to improve them?

In this paper, considering the context I thought that adding a purpose is more appropriate rather than having a definite research question. As the study is an initial step to understand construction transports therefore, it seemed to me that having a purpose is better option.

Important findings, past research gaps, and contributions to improve them have been added in the introduction chapter (line 44-62) as well as in the conclusion (line 518- 548)

Explanation:

The important findings are:

a) To identify the transport activities with respect to construction transports

b) How much time is being spent in performing value adding and non-value adding activities

These findings highlights that there is a need of better planning, communication, digitalization, use of ICT tools, better routing in order to make construction transport efficient with the focus on time.

The research gaps in the past are that there is little research being conducted within construction transports. Nevertheless, a lot of research studies have been focused on freight transports. Since the dynamics of construction transport are different than freight transports such as relatively heavy transport material, use of cranes and special equipment for loading and unloading, involvement of construction sites which are usually congested with limited space and varies geographically. Therefore, it is important to understand construction transports to bring efficiency within the construction industry.

The contributions of the research are:

- To classify construction transport activities as value adding, non-value adding and necessary but non-value adding

- To identify the time being spent in performing value adding activities and non-value adding activities

- To propose solutions to make construction transport time efficient

I would suggest the authors to draw some figures of results so that reader could clearly understand them.

Added a pie chart in section 4.3 “summary of results”

How you have improved the Construction Transport and reduce CO2 emissions, please elaborate

This has been elaborated in the conclusion (chapter 6) from line 529- 533 as well as in the discussion (chapter 5) from 475- 479.

Explanation:

By better planning, communication, coordination, digitalization, and use of ICT tools the time efficiency of construction transport will be improved. The less time will be spent on non-value adding activities such as waiting, searching for the right address to unload the material, contacting the concerned person, empty travelling etc. Likewise, there will be less CO2 emissions because unnecessary transports will get reduced. For example: during waiting, driver does not switch off the engine most of the time and as per this study, waiting can occur for 20 minutes. By better planning, communication, and coordination, by reducing waiting, CO2 emissions can be reduced.

In addition to this, moderate English language 

Thanks!

Reviewer 2 Report

Dear authors,

congratulation on the article. The study focuses on a current and really interesting topic, since the freight forwarder perspective was not always considered on the construction transport improvement. The paper is well structured, literature references are substantial,  paragraphs are smooth and well written. The choice to insert the interviews and the tables related to goods delivery flow and waste material flow as annexes is opportune.

Despite this, taking into account that your work is a pilot study and it is referred to a single case study, as interesting as it is overall, can be improved with some minor revisions:

- although rich in references and descriptions, section 2 should be extended with respect to the VSM tool considered for the identifications of value adding and non value adding, by considering other tools used in literature; at least the VSM methodology should be explained more in general. Inserting images or flow charts is recommended.

- section 3.2 and 3.3 slould contain tables, graphs ora any other graphical tool to represent the input data collected during the interviews, and the data related to the activities. The same goes for section 4.3.1 in which graphs would be more impactful than the lists of time reported.

- however well described and reasoned, the discussion and the conclusion sections do not lead to an alternative proposal to the data collected; it would be more interesting to argue about one or more improved VSM flow charts taking into account the criticalities that you found in the case study.

Warmest regards

Author Response

Dear Reviewer, 

Thank you for your time in reviewing the paper and providing very useful, relevant and insightful feedback. I have revised the paper as per your feedback. Please find attached the responses to the given comments. 

Reviewer 2 comments

Response

The VSM methodology should be explained more in general. Inserting images or flow charts is recommended.

Thanks for the suggestion. Added section on this as 2.5. Also added a figure (Figure 2.5) as an example of Value stream map.

Section 3.2 and 3.3 should contain tables, graphs or any other graphical tool to represent the input data collected during the interviews, and the data related to the activities

Table 3.3.1 and Table 3.3.2 has been added to represent the input data collected and the data related to the activities under sub section 3.3- Data Analysis of Chapter 3- Material and Method.

For section 4.3.1 in which graphs would be more impactful than the lists of time reported.

Pie chart as figure 4.3.1. and 4.3.2 has been added in section 4.3 Summary of results.

 It would be more interesting to argue about one or more improved VSM flow charts taking into account the criticalities that you found in the case study.

Added in the discussion chapter 5 as figure 5.1

Thanks! 

Round 2

Reviewer 1 Report

The paper can be accepted for publication. 

Best,